statistics/mathematical modelling

Moran process, stochastic process, birth–death process, evolutionary model, fixation time

**Author for correspondence:**
Travis Monk
e-mail: travis.monk@westernsydney.edu.au

# Martingales and the characteristic functions of absorption time on bipartite graphs

## Travis Monk and André van Schaik

International Centre for Neuromorphic Systems, The MARCS Institute, Western Sydney University, Sydney, Australia

 TM, 0000-0002-9025-0198

Evolutionary graph theory investigates how spatial constraints affect processes that model evolutionary selection, e.g. the Moran process. Its principal goals are to find the fixation probability and the conditional distributions of fixation time, and show how they are affected by different graphs that impose spatial constraints. Fixation probabilities have generated significant attention, but much less is known about the conditional time distributions, even for simple graphs. Those conditional time distributions are difficult to calculate, so we consider a close proxy to it: the number of times the mutant population size changes before absorption. We employ martingales to obtain the conditional characteristic functions (CCFs) of that proxy for the Moran process on the complete bipartite graph. We consider the Moran process on the complete bipartite graph as an absorbing random walk in two dimensions. We then extend Wald's martingale approach to sequential analysis from one dimension to two. Our expressions for the CCFs are novel, compact, exact, and their parameter dependence is explicit. We show that our CCFs closely approximate those of absorption time. Martingales provide an elegant framework to solve principal problems of evolutionary graph theory. It should be possible to extend our analysis to more complex graphs than we show here.

## 1. Introduction

The spread of some novelty in a population can be modelled by stochastic processes [1,2]. For example, stochastic processes have modelled the spread of cancer cells in healthy tissue [3,4], disease in a population [5] and social trends [6]. Birth–death processes are a subset of stochastic processes that model the spread of genetic mutations in a resident population [7–9]. One particularly popular birth–death process is the

**Figure 1.** Schematic of the Moran process on a bipartite graph and our notation. The complete bipartite graph constrains the offspring of individuals in group $A$ to replace individuals in group $B$, and vice versa. This constraint is illustrated by the black lines. All individuals are either mutants (red) or residents (blue). $S_{t-1}$ is the number of mutants in each partition on time step $t-1$, and $X_t$ is the change in the process on time step $t$. On this example time step, the offspring of a mutant from $A$ replaces a resident in $B$ (enlarged individuals and thick arrow). We repeat the Moran birth–death process until all individuals are mutants (i) or residents (ii). $\boldsymbol{a}$ and $\boldsymbol{b}$ represent the two possible final states of $S_T$.

Moran process, which has generated significant research interest since its introduction over 60 years ago [10–15].

The Moran birth–death process models the evolutionary selection of a novel mutation [10]. Briefly, it considers a population of fixed size [8]. Every individual in the population is either a mutant or a resident. On every time step, we choose one individual to reproduce and another to die. The offspring of the former replaces the latter, so the total population size remains constant. The difference between mutants and residents is that mutants have a different probability of being selected to reproduce with respect to the latter. This discrepancy is meant to model 'fitness'. We repeat this birth–death selection procedure until the entire population comprises either mutants or residents. Our goals are to find the 'fixation probability' of the initial mutant population, and the (conditional) distribution of the number of time steps required to do so [16–18].

Evolutionary graph theory studies the impact of spatial constraints on these fixation probabilities and times [11]. It considers the Moran process constrained by a graph, where the graph nodes represent individuals, and the graph edges dictate which individuals can be replaced by other individuals' offspring. The complete bipartite graph is a simple example of this concept [13,19–21]. It divides the population of individuals into two groups (figure 1). All individuals in one group are connected only to those in the other group. These connections constrain the Moran process such that the offspring from one group can only replace individuals in the other group. For example, imagine two separate colonies of sponges on the seabed. Say that the sponge larvae are programmed to swim away from their parent before searching the seabed for a suitable location to colonize [22]. If only two suitable locations exist in the immediate area, then offspring from one colony settle only in the other. We can model this simple ecosystem as a Moran process on a complete bipartite graph. We want to investigate how its spatial constraint impacts fixation probabilities and times compared with the original, fully connected Moran process [23].

The fixation probability of the complete bipartite graph is well known [19–21]. Much less is known about its conditional fixation time distributions [24–27]. Most prior work on fixation times was obtained via simulation, Markov chain analysis, restricting focus to fixation time means, or restricting population size [18,24–27]. Very few general analytical results exist for fixation times on evolutionary graphs [27], even for graphs as simple as the complete bipartite graph. In a previous paper, we showed that we can apply Wald's martingale [28] to the original, fully connected Moran process if we eliminate time steps where the mutant population size does not change [16]. We can therefore obtain tractable expressions for the full conditional characteristic functions (CCFs) of the number of times that the mutant population size changes before going extinct or fixing, i.e. its number of 'active steps' [29]. Perhaps this approach could be extended to find analogous CCFs of the Moran process on more general graphs.

In this paper, we will extend Wald's martingale to report the CCFs for the Moran process on the complete bipartite graph. We consider the bipartite graph as a two-dimensional random walk, with each dimension representing the mutant population size in one partition. We will show how to obtain fixation probabilities and the CCFs of active steps from a two-dimensional product martingale. Our expressions for the CCFs are novel, elegant, exact, and their parameter dependence is explicit. We will

investigate the parameter dependence of our CCFs by evaluating them in different regions of parameter space. We will show that our CCFs of the number of active steps can accurately approximate the CCFs of fixation time. We will establish conditions for that approximation to be particularly accurate. Our analysis demonstrates that martingales are a powerful tool to solve fundamental problems in evolutionary graph theory, often within a few lines of mathematics [16,21,30–32].

# 2. Results

## 2.1. Problem statement and notation

For a more detailed introduction to the Moran birth–death process, see [8,11].

Figure 1 is a schematic of the Moran birth–death process on a complete bipartite graph [19,20,33]. All individuals in a population are divided into two partitions of sizes $A$ and $B$, e.g. $A = 4$ and $B = 2$ in figure 1. Individuals are either mutants (red nodes) or residents (blue nodes). The only difference between the species is that mutants are chosen to reproduce with a different probability relative to the residents. This difference is meant to model 'fitness' and is parametrized by $r$ [8]. All individuals in one partition are connected only to those in the other partition (black lines, figure 1). These connections constrain the Moran process such that offspring from one partition can only replace individuals in the other partition.

We consider the Moran process on the complete bipartite graph as a two-dimensional random walk, where the mutant population size fluctuates in two partitions. Let $S_{t-1} = [S_{a,t-1}, S_{b,t-1}]$ represent the size of the mutant populations in each partition on time step $t - 1$ (e.g. $S_{t-1} = [2, 1]$ in figure 1). Let $X_t = [X_{a,t}, X_{b,t}]$ represent the change in the mutant population size on time step $t$. For example, if a mutant from $A$ is chosen to reproduce and a resident from $B$ to die, then $X_t = [0, 1]$ (enlarged individuals and thick connection, figure 1). We write $S_t = \sum_{i=1}^{t} X_i + S_0$, where $S_0 = [S_{a,0}, S_{b,0}]$ is the initial mutant population size. On every time step we make a new observation of $X_t$ and add it to the sum until all individuals are mutants (figure 1b(i)) or residents (b(ii) graph).

Our goal is to find the probability that an initial mutant population $S_0$ eventually achieves fixation (i.e. the fixation probability), and how many time steps $T$ are required to do so. Since all connections are undirected, $T$ is almost surely finite [34]. Let $a = [A, B]$ and $b = [0, 0]$ represent the two possible final states of the bipartite graph (figure 1b). The fixation probability is then $\Pr(S_T = a) \equiv \alpha$ and the extinction probability is $\Pr(S_T = b) = 1 - \alpha$. We also want to find the conditional distributions $\Pr(T = t | S_T = a)_{t=0}^{\infty}$ and $\Pr(T = t | S_T = b)_{t=0}^{\infty}$ of $T$.

It is very difficult to calculate $\Pr(T = t | S_T = a)_{t=0}^{\infty}$ and $\Pr(T = t | S_T = b)_{t=0}^{\infty}$, even for simpler birth–death processes like the fully connected, one-dimensional Moran process [16–18]. Instead, we consider the number of times that the mutant population size has changed upon absorption $C_T$. Let $Y_t$ represent whether the mutant population size changes on time step $t$

$$Y_t = 1 \text{ if } X_t \neq [0, 0]; \quad Y_t = 0 \text{ if } X_t = [0, 0].$$

Initializing $C_0 = 0$, we write $C_T = \sum_{i=1}^{T} Y_i$. Note that $C_T$ depends on $T$, so we interpret them as proxies to each other [16].

We will identify a product martingale that yields $\alpha$ and the full CCFs of $C_T$.

## 2.2. Fixation probability and times from a two-dimensional martingale

First, we show how to obtain the fixation probability and CCFs of $C_T$ from a two-dimensional product martingale, assuming that we can find one. Say we find a product martingale of the form

$$\mathbb{E}[h^{C_t} f^{S_{a,t}} g^{S_{b,t}} | S_{t-1}] = h^{C_{t-1}} f^{S_{a,t-1}} g^{S_{b,t-1}}, \tag{2.1}$$

where $h$ is a free complex variable and $f = f(h)$ and $g = g(h)$ are functions of $h$ that are independent of $S_{t-1}$. We say that $h$, $f(h)$ and $g(h)$ satisfying equation (2.1) define a 'product martingale' because its exponentiation turns sums into products

$$\mathbb{E}[h^{C_t} f^{S_{a,t}} g^{S_{b,t}} | S_{t-1}] = \mathbb{E}[h^{C_{t-1}+Y_t} f^{S_{a,t-1}+X_{a,t}} g^{S_{b,t-1}+X_{b,t}} | S_{t-1}]$$

$$= h^{C_{t-1}} f^{S_{a,t-1}} g^{S_{b,t-1}} \mathbb{E}[h^{Y_t} f^{X_{a,t}} g^{X_{b,t}} | S_{t-1}].$$

If we can show that $\mathbb{E}[h^{Y_t} f^{X_{a,t}} g^{X_{b,t}} | S_{t-1}] = 1$, then equation (2.1) is true and we have a product martingale.

We can immediately calculate the fixation probability and CCFs of $C_T$ from equation (2.1) [28]. Taking the expectation of both sides of equation (2.1)

$$\mathbb{E}[h^{C_t}f^{S_{a,t}}g^{S_{b,t}}] = \mathbb{E}[h^{C_{t-1}}f^{S_{a,t-1}}g^{S_{b,t-1}}].$$

By induction

$$\mathbb{E}[h^{C_t}f^{S_{a,t}}g^{S_{b,t}}] = \mathbb{E}[h^{C_0}f^{S_{a,0}}g^{S_{b,0}}] = f^{S_{a,0}}g^{S_{b,0}},$$

assuming that $S_0$ is known (non-random) and $C_0 = 0$. Doob's optional stopping theorem states that a randomly stopped martingale is also a martingale [35,36]. Inserting a random variable $T$ for $t$

$$\mathbb{E}[h^{C_T}f^{S_{a,T}}g^{S_{b,T}}] = f^{S_{a,0}}g^{S_{b,0}}.$$

Splitting the expectation, conditional on fixation or extinction

$$\mathbb{E}[h^{C_T}f^{S_{a,T}}g^{S_{b,T}}|\mathbf{S_T} = \mathbf{a}]\alpha + \mathbb{E}[h^{C_T}f^{S_{a,T}}g^{S_{b,T}}|\mathbf{S_T} = \mathbf{b}](1-\alpha) = f^{S_{a,0}}g^{S_{b,0}}.$$

Inserting the fixation and extinction boundaries

$$f^A g^B \mathbb{E}[h^{C_T}|\mathbf{S_T} = \mathbf{a}]\alpha + \mathbb{E}[h^{C_T}|\mathbf{S_T} = \mathbf{b}](1-\alpha) = f^{S_{a,0}}g^{S_{b,0}}. \tag{2.2}$$

We obtain the fixation probability and times from equation (2.2) by inserting special values for the free variable $h$ into it [16]. For the fixation probability, insert $h = 1$ (recall that $f$ and $g$ are functions of $h$)

$$f(1)^A g(1)^B \alpha + (1-\alpha) = f(1)^{S_{a,0}}g(1)^{S_{b,0}}.$$

Rearranging for $\alpha$

$$\alpha = \frac{f(1)^{S_{a,0}}g(1)^{S_{b,0}} - 1}{f(1)^A g(1)^B - 1}.$$

For the CCFs, insert $h = e^\tau$ into equation (2.2), where $\tau$ is a purely imaginary free variable

$$f(e^\tau)^A g(e^\tau)^B \mathbb{E}[e^{\tau C_T}|\mathbf{S_T} = \mathbf{a}]\alpha + \mathbb{E}[e^{\tau C_T}|\mathbf{S_T} = \mathbf{b}](1-\alpha) = f(e^\tau)^{S_{a,0}}g(e^\tau)^{S_{b,0}}.$$

We recognize the conditional expectations as the CCFs of $C_T$, $\psi_{C_T|\mathbf{S_T}=\mathbf{a}}(\tau)$ and $\psi_{C_T|\mathbf{S_T}=\mathbf{b}}(\tau)$

$$f(e^\tau)^A g(e^\tau)^B \psi_{C_T|\mathbf{a}}(\tau)\alpha + \psi_{C_T|\mathbf{b}}(\tau)(1-\alpha) = f(e^\tau)^{S_{a,0}}g(e^\tau)^{S_{b,0}}.$$

Assume that there are two pairs of complex functions $(f_1(h), g_1(h))$ and $(f_2(h), g_2(h))$ that satisfy equation (2.1) [28]. Separately inserting those pairs into equation (2.2), we obtain a system of two equations

and

$$\left.\begin{array}{l} f_1(e^\tau)^A g_1(e^\tau)^B \psi_{C_T|\mathbf{a}}(\tau)\alpha + \psi_{C_T|\mathbf{b}}(\tau)(1-\alpha) = f_1(e^\tau)^{S_{a,0}}g_1(e^\tau)^{S_{b,0}} \\ f_2(e^\tau)^A g_2(e^\tau)^B \psi_{C_T|\mathbf{a}}(\tau)\alpha + \psi_{C_T|\mathbf{b}}(\tau)(1-\alpha) = f_2(e^\tau)^{S_{a,0}}g_2(e^\tau)^{S_{b,0}}. \end{array}\right\} \tag{2.3}$$

We have two equations, so we can solve for both $\psi_{C_T|\mathbf{a}}(\tau)$ and $\psi_{C_T|\mathbf{b}}(\tau)$.

The key condition that we need to apply this analysis is

$$\mathbb{E}[h^{Y_t}f^{X_{a,t}}g^{X_{b,t}}|\mathbf{S_{t-1}}] = 1, \tag{2.4}$$

for two pairs of *state-independent* $f(h)$ and $g(h)$. We now show that this condition can be met for the Moran process on a complete bipartite graph.

## 2.3. A two-dimensional martingale for the complete bipartite graph

For compact notation, let $F_{t-1}$ represent the total fitness of the bipartite graph on time step $t-1$: $F_{t-1} = rS_{a,t-1} + A - S_{a,t-1} + rS_{b,t-1} + B - S_{b,t-1}$. We use shorthand for the graph's transition probabilities

$$\begin{array}{ll} p_{X_a\uparrow} = \Pr(\mathbf{X_t} = [1, 0], Y_t = 1|\mathbf{S_{t-1}}), & p_{X_a\downarrow} = \Pr(\mathbf{X_t} = [-1, 0], Y_t = 1|\mathbf{S_{t-1}}); \\ p_{X_b\uparrow} = \Pr(\mathbf{X_t} = [0, 1], Y_t = 1|\mathbf{S_{t-1}}), & p_{X_b\downarrow} = \Pr(\mathbf{X_t} = [0, -1], Y_t = 1|\mathbf{S_{t-1}}); \\ p_{X0} = \Pr(\mathbf{X_t} = [0, 0], Y_t = 0|\mathbf{S_{t-1}}). & \end{array}$$

These transition probabilities are

$$p_{X_a\uparrow} = \frac{rS_{b,t-1}}{F_{t-1}}\frac{A - S_{a,t-1}}{A}, \qquad p_{X_a\downarrow} = \frac{B - S_{b,t-1}}{F_{t-1}}\frac{S_{a,t-1}}{A};$$

$$p_{X_b\uparrow} = \frac{rS_{a,t-1}}{F_{t-1}}\frac{B - S_{b,t-1}}{B}, \qquad p_{X_b\downarrow} = \frac{A - S_{a,t-1}}{F_{t-1}}\frac{S_{b,t-1}}{B};$$

$$p_{X0} = 1 - p_{X_a\uparrow} - p_{X_b\uparrow} - p_{X_a\downarrow} - p_{X_b\downarrow}.$$

We want to find state-independent $f(h)$ and $g(h)$ such that equation (2.4) is true. Writing the expectation

$$\mathbb{E}[h^{Y_t} f^{X_{a,t}} g^{X_{b,t}} | \mathbf{S_{t-1}}] = p_{X_a\uparrow}hf + p_{X_a\downarrow}hf^{-1} + p_{X_b\uparrow}hg + p_{X_b\downarrow}hg^{-1} + p_{X0} = 1.$$

Inserting $p_{X0}$ and rearranging

$$p_{X_a\uparrow}hf + p_{X_a\downarrow}hf^{-1} + p_{X_b\uparrow}hg + p_{X_b\downarrow}hg^{-1} = p_{X_a\uparrow} + p_{X_a\downarrow} + p_{X_b\uparrow} + p_{X_b\downarrow}. \qquad (2.5)$$

Equation (2.5) is true if the following two equations are true

$$p_{X_a\uparrow}hf + p_{X_b\downarrow}hg^{-1} = p_{X_a\uparrow} + p_{X_b\downarrow} \quad \text{and} \quad p_{X_a\downarrow}hf^{-1} + p_{X_b\uparrow}hg = p_{X_a\downarrow} + p_{X_b\uparrow}.$$

We split equation (2.5) this way because when we insert our expressions for transition probabilities, all state dependence cancels

$$\frac{r}{A}hf + \frac{1}{B}hg^{-1} = \frac{r}{A} + \frac{1}{B} \quad \text{and} \quad \frac{r}{B}hg + \frac{1}{A}hf^{-1} = \frac{r}{B} + \frac{1}{A}.$$

With two equations, we can solve for $f$ and $g$ as functions of $h$. Rearranging the right equation for $g$

$$g = \frac{1}{h} + \frac{B}{Arh} - \frac{B}{Ar}f^{-1}.$$

Substituting $g$ in the left equation and rearranging, we see that $f$ is the solution to a quadratic equation

$$\left(\frac{r}{A} + \frac{B}{A^2}\right)f^2 + \left(\frac{h}{B} - \frac{Bh}{A^2} - \left(\frac{r}{A} + \frac{1}{B}\right)\left(\frac{1}{h} + \frac{B}{Arh}\right)\right)f + \left(\frac{r}{A} + \frac{1}{B}\right) = 0.$$

There are two pairs of complex functions $(f_1(h), g_1(h))$ and $(f_2(h), g_2(h))$ that satisfy equation (2.4) for the Moran process on the complete bipartite graph. In particular, $f_1$ and $f_2$ are given by the quadratic formula (one corresponds to the plus sign and the other to the minus sign in the quadratic formula). Then $g_1$ and $g_2$ are linearly related to the inverse of those two solutions.

Figure 2 plots $f_1$, $f_2$ (a,c), $g_1$ and $g_2$ (b,d) as functions of $\tau$, where $h = e^\tau$. We plot these functions for $r = 0.5$ (a,b) and $r = 1.5$ (c,d). The real (red traces) and imaginary (black traces) parts of each function are plotted separately. Note that the real and imaginary parts of all functions are even and odd about $\tau = 0$ respectively. Characteristic functions also have this property. Each panel in figure 2 shows that there are two functions $f_1$ (solid traces, a,c) and $f_2$ (dashed traces, a,c), and $g_1$ (solid traces, b,d) and $g_2$ (dashed traces, b,d) that satisfy equation (2.4).

When $\tau = 0$, one of those two functions passes through the point $(1, 0i)$ (pink and grey dots, figure 2). This observation reflects the fact that, when $h = 1$ (or $\tau = 0$), the solution $f = 1$ and $g = 1$ to equation (2.4) is trivial. The other function passes through a non-trivial point $(f_0, 0i)$ or $(g_0, 0i)$ when $\tau = 0$ (red and black dots, figure 2). These points are what we use to evaluate the fixation probability.

Setting $h = 1$ (or $\tau = 0$) and solving for $f$ and $g$ above, we find those non-trivial points:

$$f_0 = \frac{A + Br}{r(Ar + B)} \quad \text{and} \quad g_0 = \frac{Ar + B}{r(A + Br)}.$$

The fixation probability of the bipartite graph is then

$$\alpha = \frac{f_0^{S_{a,0}} g_0^{S_{b,0}} - 1}{f_0^A g_0^B - 1},$$

which is consistent with previous results [20,21,33].

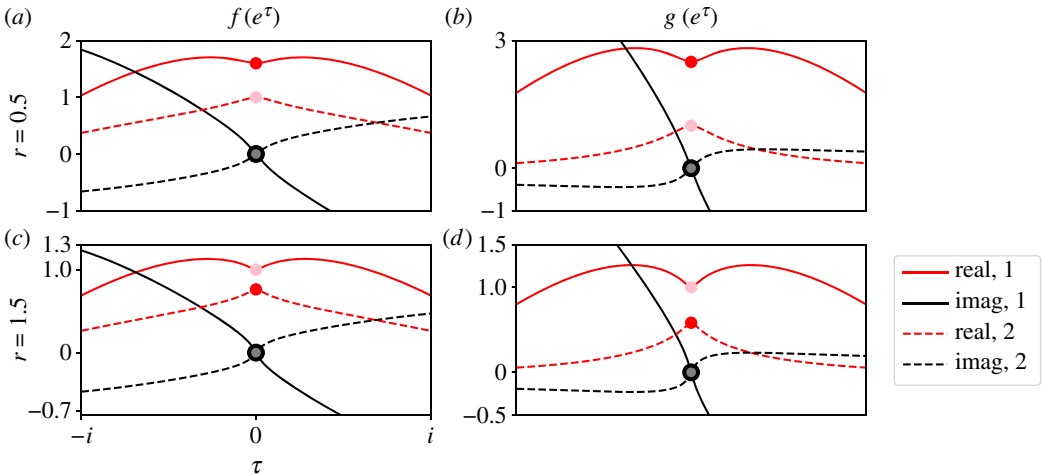

**Figure 2.** Two pairs of state-independent functions ($f_1$, $g_1$) and ($f_2$, $g_2$) satisfy equation (2.4) for the Moran process on the complete bipartite graph. Those functions are plotted for example values of $r = 0.5$ ($a,b$) and $r = 1.5$ ($c,d$). Panel ($a,c$) plots $f_1$ (solid traces) and $f_2$ (dashed traces) while ($b,d$) plots $g_1$ (solid traces) and $g_2$ (dashed traces). Real (red) and imaginary (black) parts of all functions are plotted separately. Note that the real parts are even and the imaginary parts are odd. When $\tau = 0$, one function passes through ($1$, $0i$) (pink, grey dots) and the other passes through ($f_0$, $0i$) or ($g_0$, $0i$) (red, black dots). The red dots are the values we use to calculate the fixation probability.

We obtain the CCFs of $C_T$ by rearranging equation (2.3)

$$\psi_{C_T|a}(\tau) = \frac{f_1^{S_{a,0}} g_1^{S_{b,0}} - f_2^{S_{a,0}} g_2^{S_{b,0}}}{\alpha(f_1^A g_1^B - f_2^A g_2^B)} \quad \text{and} \quad \psi_{C_T|b}(\tau) = \frac{f_2^{S_{a,0}} g_2^{S_{b,0}} f_1^A g_1^B - f_1^{S_{a,0}} g_1^{S_{b,0}} f_2^A g_2^B}{(1-\alpha)(f_1^A g_1^B - f_2^A g_2^B)},$$

and inserting $h = e^{\tau}$ in our expressions for $f_1$, $f_2$, $g_1$ and $g_2$.

In the special case $A = B$ (i.e. the graph is isothermal), those functions have a simple form

$$f_1 = g_1 = \frac{(r+1)e^{-\tau} + \sqrt{(r+1)^2 e^{-2\tau} - 4r}}{2r} \quad \text{and} \quad f_2 = g_2 = \frac{(r+1)e^{-\tau} - \sqrt{(r+1)^2 e^{-2\tau} - 4r}}{2r}.$$

Since $f_1 = g_1$ and $f_2 = g_2$, our two-dimensional product martingale reduces to one dimension. Furthermore, $f_1$ and $f_2$ are equivalent to functions derived from a one-dimensional product martingale applied to the fully connected Moran process (c.f. eqn. 2.7 in [16]). When the bipartite graph is isothermal, its CCFs of $C_T$ are equivalent to those of the fully connected Moran process [16].

## 2.4. Parameter dependence of the CCFs

The parameter dependence of $\psi_{C_T|b}(\tau)$ and $\psi_{C_T|a}(\tau)$ is explicit, so we can investigate their parameter dependence by simply evaluating them in different regions of parameter space.

### 2.4.1. Strong selection expedites extinction and fixation

Figure 3 plots $\psi_{C_T|b}(\tau)$ ($a,c$) and $\psi_{C_T|a}(\tau)$ ($b,d$) for a complete bipartite graph with two values of $r$ ($a,b$ and $c$, $d$). Our parameter values were $A = 10$, $B = 4$, $S_{a,0} = 1$ and $S_{b,0} = 1$. We plot the real (pink) and imaginary (grey) parts of the CCFs separately. Note that the real and imaginary parts of the CCFs are even and odd, and they pass through 1 and 0 at $\tau = 0$, respectively.

Figure 3 compares $\psi_{C_T|b}(\tau)$ and $\psi_{C_T|a}(\tau)$ (solid traces) with simulation results from 100 000 trials of the Moran process (dashed traces). On each trial, we counted how many times the mutant population size changed, and whether it fixed or went extinct. We then applied the Fourier transform to that simulation data, and again plot its real and imaginary parts separately (dashed red and black traces, figure 3). We also compared our expression for $\alpha$ with the percentage of simulations where the mutants fixed (lower-right numbers, $c,d$). Our simulation code is available online at https://github.com/travismonk/bipartite. Simulation results match our theory extremely closely because our analysis is exact, and we ran sufficiently many simulations to converge to that solution.

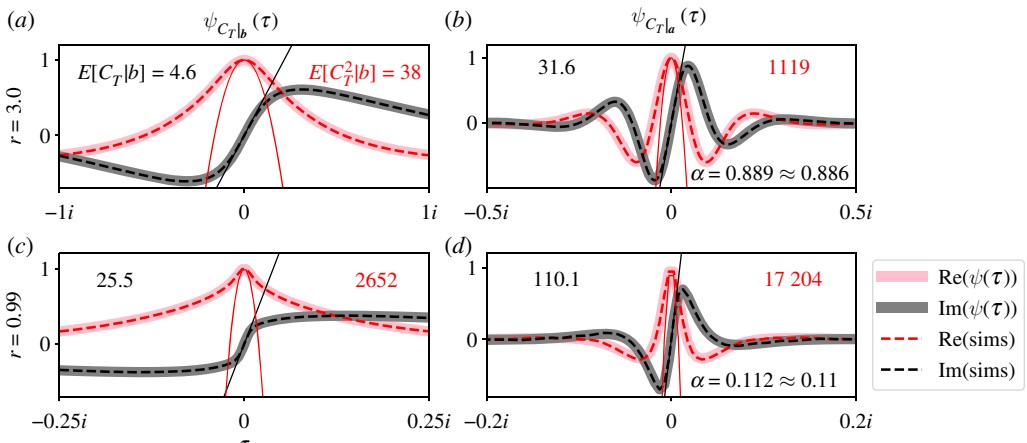

**Figure 3.** Exact CCFs of $C_T$ for the Moran process on the complete bipartite graph. $\psi_{C_T|b}(\tau)$ (a,c) and $\psi_{C_T|a}(\tau)$ (b,d) for two values of $r$ (a,b and c,d). Real (solid pink) and imaginary (solid grey) parts of the CCFs are plotted separately. Fourier transforms of 100 000 simulations (dashed traces) match our theoretical CCFs, as do the percentage of successful invasions with our expression for $\alpha$ (lower-right corners of right panels). The black and red numbers in each panel report the conditional first and second moments of $C_T$, respectively. The black line and red parabola visualize those two moments. In all plots, $A = 10$, $B = 4$, $S_{a,0} = 1$ and $S_{b,0} = 1$.

Figure 3 also reports the conditional first two moments of $\psi_{C_T|b}(\tau)$ and $\psi_{C_T|a}(\tau)$. The black and red numbers in the left panels report $\mathbb{E}[C_T|\mathbf{S_T} = b]$ and $\mathbb{E}[C_T^2|\mathbf{S_T} = b]$, respectively. They report $\mathbb{E}[C_T|\mathbf{S_T} = a]$ and $\mathbb{E}[C_T^2|\mathbf{S_T} = a]$ in the right panels. Those moments are visualized by the black line and red parabola in each panel of figure 3. Analytical expressions for the conditional $k$th moment are found by evaluating derivatives of the CCFs

$$\mathbb{E}[C_T^k|\mathbf{S_T} = a] = i^{-k}\frac{\mathrm{d}^k}{\mathrm{d}\tau^k}\psi_{C_T|a}(\tau)\Big|_{\tau=0} \quad \text{and} \quad \mathbb{E}[C_T^k|\mathbf{S_T} = b] = i^{-k}\frac{\mathrm{d}^k}{\mathrm{d}\tau^k}\psi_{C_T|b}(\tau)\Big|_{\tau=0}.$$

Those analytical expressions are not compact, so we omit them here. But it is easy to estimate at least the first few moments by visually inspecting the CCFs (e.g. black lines and red parabolas, figure 3). We can investigate how those moments depend on parameters by visually comparing CCFs that we calculate in different regions of parameter space.

For example, figure 3 shows that the first and second moments of $C_T$ decrease as selection increases (cf. a,b and c,d). When $r$ is large, the drift of the mutant population size is more positive. When the drift is strongly positive, $S_t$ is unlikely to increase far from $S_0$ and then go extinct. Such paths to extinction require larger $C_T$ on average because they traverse more states of the graph. So it is less likely that a path with large $C_T$ will go extinct, because many of those paths are very unlikely to be observed. Therefore, when $r$ is large, extinctions usually happen quickly. By the same logic, when $r$ is large, it is unlikely for $S_t$ to decrease substantially before reaching $a$. Those paths to extinction also require larger $C_T$ on average, so fixation usually happens quickly once the initial mutants gain a foothold on the graph. These arguments are no longer true when $r = 1$. When selection is neutral, it is more possible for $S_t$ to drift higher from $S_0$ and then go extinct, or drift lower and then fix. Since these longer paths to fixation or extinction are more probable, the conditional first and second moments of $C_T$ increase (cf. black and red numbers, figure 3).

### 2.4.2. Increasing the population size delays fixation more than extinction

Figure 4 illustrates how $\psi_{C_T|b}(\tau)$ (a) and $\psi_{C_T|a}(\tau)$ (b) depend on the total population size of the complete bipartite graph (i.e. $A + B$). Again, we plot the real (red/pink traces) and imaginary (black/grey traces) parts of the CCFs separately. We fixed $r = 1.01$, $S_{a,0} = 1$ and $S_{b,0} = 1$, and calculated CCFs of $C_T$ for three pairs of values for $A$ and $B$. In all plots, we constrained the population sizes $A$ and $B$ to have the same ratio $A/B = 3$ (figure 4).

Figure 4 shows that $\psi_{C_T|a}(\tau)$ is more sensitive to population size than $\psi_{C_T|b}(\tau)$. $\psi_{C_T|a}(\tau)$ concentrates in a smaller neighbourhood about the origin as population size grows. That concentration dramatically affects its second moment (red numbers, b), as illustrated by the dashed red or pink parabolas. For these parameter values, that second moment increases by over an order of magnitude when the population size doubles. $\psi_{C_T|b}(\tau)$ is also dependent on $a$, but not as dramatically as $\psi_{C_T|a}(\tau)$. The mean

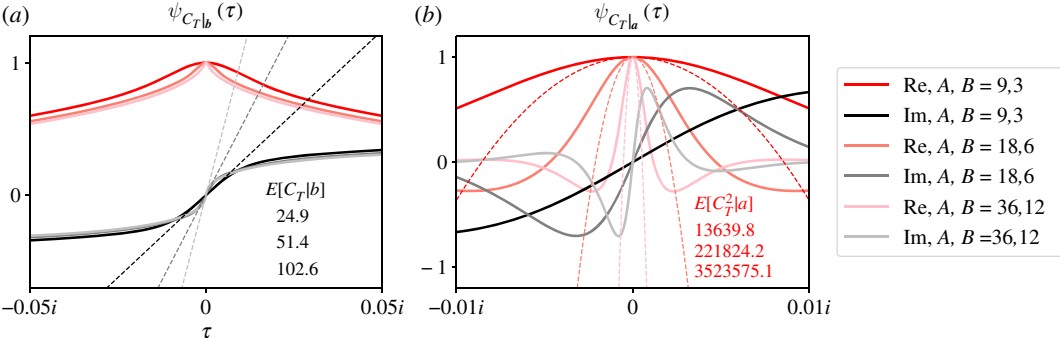

**Figure 4.** Increasing the population size moderately delays extinction and significantly delays fixation. $\psi_{C_T|b}(\tau)$ (a) and $\psi_{C_T|a}(\tau)$ (b) are plotted for three bipartite graphs with different partition sizes as stated in the legend. All bipartite graphs have three times more individuals in one partition than the other. Again, real (red/pink) and imaginary (black/grey) parts of the CCFs are plotted separately. The top, middle and bottom numbers report particular conditional moments of $C_T$ for the bipartite graph with partition sizes $(A, B) = (9, 3)$, $(18, 6)$ and $(36, 12)$, respectively. As population size increases, both conditional means of $C_T$ increase. The slopes of the lines in a illustrate this increase for $C_T|b$, and the black numbers report those slopes. Both second moments of $C_T$ increase as well, as illustrated by the red parabolas and red numbers in b. Inspecting the CCFs away from the origin, we see that the higher-order moments of $C_T|a$ are more sensitive to changes in population size than $C_T|b$. In these plots, $r = 1.01$, $S_{a,0} = 1$ and $S_{b,0} = 1$.

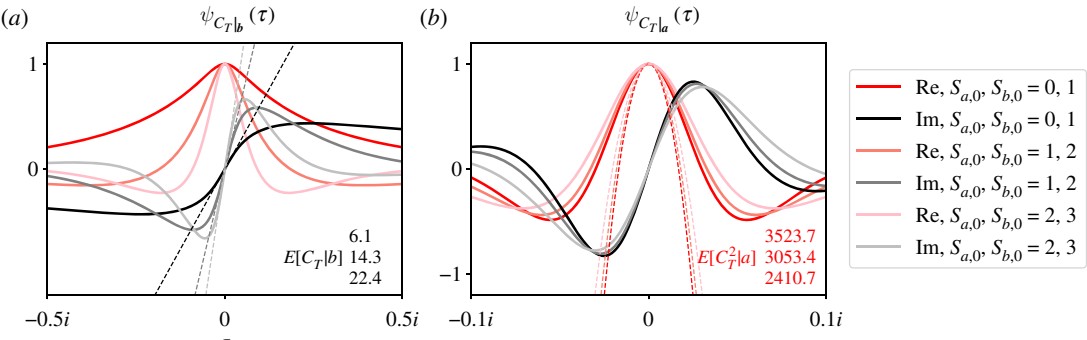

**Figure 5.** Increasing the initial mutant population size delays extinction and slightly expedites fixation. The figure layout is analogous to figure 4. As we increase $S_0$, we increase the distance between $S_0$ and $b$, and decrease the distance between $S_0$ and $a$. Consequently, $\psi_{C_T|b}(\tau)$ condenses about the origin, and $\psi_{C_T|a}(\tau)$ expands from the origin. Equivalently, the moments of $C_T|b$ and $C_T|a$ increase and decrease, respectively. When $S_0$ is small compared with the population size, we see that $\psi_{C_T|b}(\tau)$ is more sensitive to changes in $S_0$ than $\psi_{C_T|a}(\tau)$. In these plots, $r = 2$, $A = 3$ and $B = 10$.

of $C_T$ to extinction (black numbers, a) doubles as population size doubles, as visualized by the slopes of the dashed lines. Higher-order moments of $C_T|b$ do not appear to be sensitive to changes in $a$.

These results are qualitatively similar to previous results we found for the fully connected Moran process [16]. As $a$ moves farther from $S_0$, $\mathbb{E}[C_T|a]$ increases because $S_t$ must traverse a larger number of states, which implies a larger number of population size changes. But $\mathbb{E}[C_T|b]$ increases as well because longer paths to extinction become possible. For example, if $a = (9, 3)$, then the Moran process cannot visit the state $(9, 3)$ and then go extinct because it is already fixed. If we increase $a$, then that long path to extinction becomes possible. $\mathbb{E}[C_T^2|b]$ and $\mathbb{E}[C_T^2|a]$ increase as $a$ increases for the same reason. As the distance between $b$ and $a$ increases, we can observe longer paths to absorption. Summing the square of those longer path lengths can result in a significantly higher second moment of $C_T$. This observation is especially true when the probabilities of observing those longer paths are non-negligible, i.e. when selection is weak.

### 2.4.3. Increasing the starting state delays extinction more than fixation

Figure 5 is analogous to figure 4, except we altered the starting state $S_0$ instead of $a$. We fixed $r = 2$, $A = 3$, $B = 10$, and varied $S_0$ as indicated in the legend. Figure 5 shows that $\psi_{C_T|b}(\tau)$ is more sensitive to changes

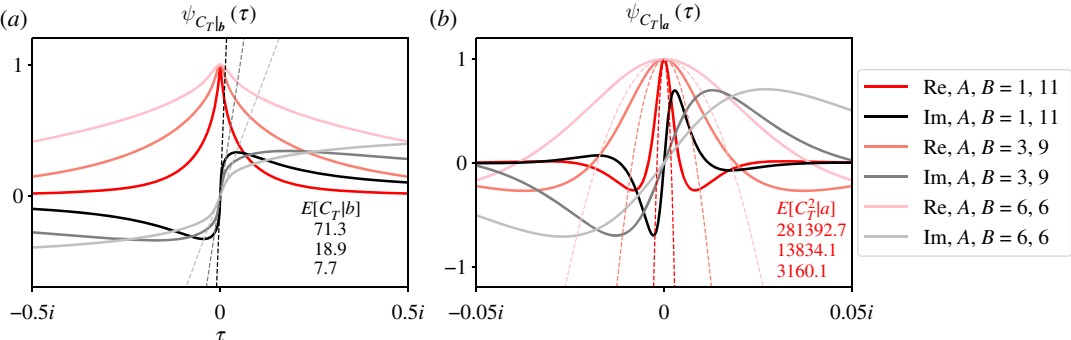

**Figure 6.** Asymmetrically distributing individuals between partitions delays extinction and fixation. The figure layout is analogous to figures 4 and 5. As we increase the asymmetry of partition sizes, both CCFs concentrate about the origin. Equivalently, the moments of $C_T|\boldsymbol{b}$ and $C_T|\boldsymbol{a}$ increase, as illustrated by the black and red numbers. The complete bipartite graph may amplify the fixation probability of an advantageous mutation [21], but it also requires more mutant population size changes to achieve fixation. This result is consistent with previous work on fixation time [26]. In these plots, $r = 1.01$, $S_{a,0} = 0$ and $S_{b,0} = 1$.

in $S_0$ than $\psi_{C_T|\boldsymbol{a}}(\tau)$ when $S_0$ is closer to $\boldsymbol{b}$ than $\boldsymbol{a}$. For example, increasing $S_{a,0}$ and $S_{b,0}$ by one more than doubles $\mathbb{E}[C_T|\boldsymbol{b}]$ (black numbers, dashed black/grey lines, left panel). By contrast, $\mathbb{E}[C_T|\boldsymbol{a}]$ and $\mathbb{E}[C_T^2|\boldsymbol{a}]$ decrease slightly as $S_0$ increases.

These results are sensible. As we increase the distance between $S_0$ and $\boldsymbol{b}$, we expect $C_T|\boldsymbol{b}$ to increase because there are more states for $S_t$ to traverse before extinction. Since increasing the distance between $S_0$ and $\boldsymbol{b}$ necessarily decreases the distance between $S_0$ and $\boldsymbol{a}$, we expect $C_T|\boldsymbol{a}$ to decrease for the converse reason. However, increasing $S_0$ also makes longer paths to fixation possible. For example, if $S_0 = (S_{a,0}, S_{b,0}) = (1, 0)$, then $S_{a,t}$ cannot decrease by 1 and then fix because the mutants already went extinct. But if $S_0 = (2, 0)$, then this path to fixation is possible, and it requires a slightly larger number of mutant population size changes on average. These two effects of increasing $S_0$ on $C_T|\boldsymbol{a}$ partially offset each other, so $\psi_{C_T|\boldsymbol{a}}(\tau)$ is relatively insensitive to changes in $S_0$.

### 2.4.4. Asymmetric partition sizes delays extinction and fixation

Figure 6 is analogous to figures 4 and 5, except we fixed the total population size in all plots. We fixed $r = 1.01$, $S_{a,0} = 0$ and $S_{b,0} = 1$, and calculated the CCFs for three pairs of values for $A$ and $B$. In all plots, we constrained $A$ and $B$ to sum to 12, but placed different numbers of those 12 individuals in the partitions (figure 6 legend). By doing so we can investigate how $\psi_{C_T|\boldsymbol{b}}(\tau)$ and $\psi_{C_T|\boldsymbol{a}}(\tau)$ depend on the asymmetry of partition sizes. We can also investigate how this asymmetry impacts the CCFs with respect to an isothermal graph.

Figure 6 shows that asymmetric partition sizes significantly impact both CCFs. It shows that the isothermal graph has the lowest first and second moments of both $C_T|\boldsymbol{b}$ and $C_T|\boldsymbol{a}$ (lightest pink and grey traces). As the population sizes of the partitions become increasingly asymmetric, the graph requires more population size changes to achieve extinction or fixation. To explain this observation, consider a star graph [11–13,21,25], i.e. a bipartite graph where one partition has only one individual and the other has many individuals (red and black traces, figure 6). On a given time step, an individual from the populous partition is more likely to be selected to reproduce because that partition has more individuals. Then the lonely individual in the other partition will be replaced on most time steps. Therefore, most mutant population size changes are that lonely individual flipping between mutant and resident. So the star graph requires many population size changes for the mutant population size in the populous partition to grow or shrink. This result is consistent with previous computational simulations showing that the fixation time $T$ increases as the asymmetry of partition sizes increases [26].

## 2.5. Approximating the CCFs of the number of time steps

Figure 7 compares the CCFs of $C_T$ with those of $T$. Figure 7a,c compares $\psi_{C_T|\boldsymbol{b}}(\tau)$ (solid traces) with $\psi_{T|\boldsymbol{b}}(\tau)$ (dashed traces), and figure 7b,d compares $\psi_{C_T|\boldsymbol{a}}(\tau)$ with $\psi_{T|\boldsymbol{a}}(\tau)$. The bottom and top x-axes in figure 7 correspond to the independent variable of the CCFs of $C_T$ and $T$, respectively. Again, the real (pink

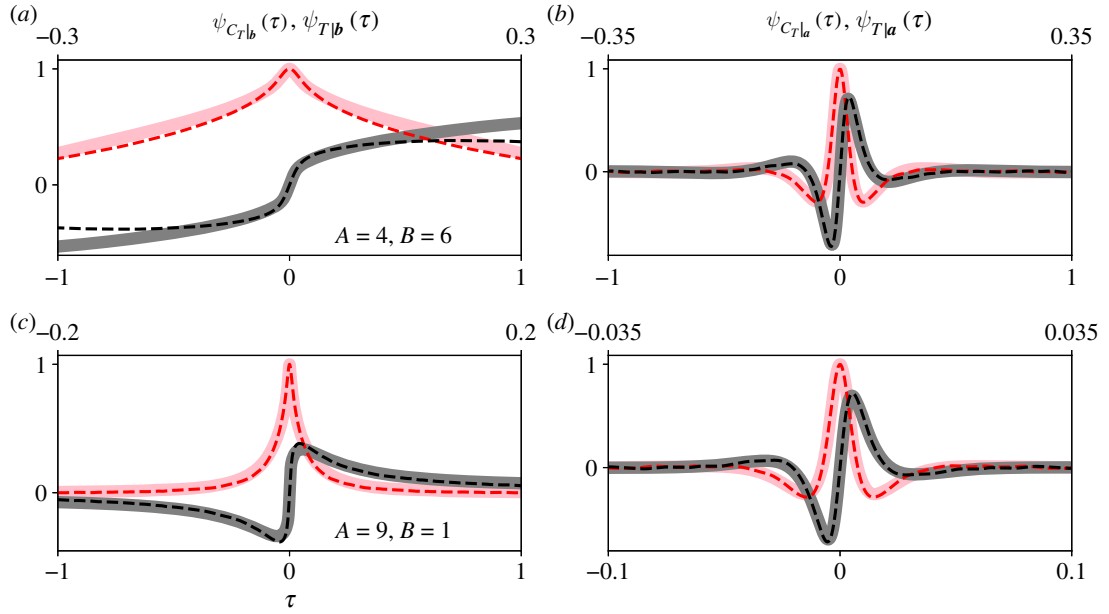

**Figure 7.** Our CCFs of $C_T$ approximate those of $T$ when the population size is small and selection is weak, or when partition sizes are asymmetric. $\psi_{C_T|\boldsymbol{b}}(\tau)$ (solid traces, $a,c$) and $\psi_{C_T|\boldsymbol{a}}(\tau)$ (solid traces, $b,d$) approximate $\psi_{T|\boldsymbol{b}}(\tau)$ (dashed traces, $a,c$) and $\psi_{T|\boldsymbol{a}}(\tau)$ (dashed traces, $b,d$) to within a scaling constant. We obtained $\psi_{T|\boldsymbol{b}}(\tau)$ and $\psi_{T|\boldsymbol{a}}(\tau)$ from 200 000 simulations. In all panels, $\psi_{C_T|\boldsymbol{b}}(\tau)$ and $\psi_{C_T|\boldsymbol{a}}(\tau)$ are plotted with respect to the bottom $x$-axis, and $\psi_{T|\boldsymbol{b}}(\tau)$ and $\psi_{T|\boldsymbol{a}}(\tau)$ are plotted with respect to the top $x$-axis. The real (pink and red) and imaginary (grey and black) parts of the CCFs are plotted separately. Note that, when the partition sizes are highly asymmetric ($c,d$), both the fixation and extinction CCFs closely match each other to within a scaling constant. When the partition sizes are approximately equal ($a,b$), the higher-order moments of $C_T|\boldsymbol{b}$ do not approximate those of $T|\boldsymbol{b}$ as accurately. These observations are analogous to those made for the fully connected Moran process [16]. In all plots, $r = 1.01$, $S_{a,0} = 1$ and $S_{b,0} = 0$.

and red) and imaginary (grey and black) parts of the CCFs are plotted separately. We obtained $\psi_{T|\boldsymbol{a}}(\tau)$ and $\psi_{T|\boldsymbol{b}}(\tau)$ by simulating the Moran process on the complete bipartite graph 200 000 times. Our simulation code is available online at https://github.com/travismonk/bipartite. We stored $T|\boldsymbol{a}$ or $T|\boldsymbol{b}$ after each simulation, depending on whether the Moran process achieved fixation or extinction, and computed their Fourier transforms. In all plots, $r = 1.01$, $S_{a,0} = 1$ and $S_{b,0} = 0$. The partition sizes were either $A = 4$ and $B = 6$ ($a,b$) or $A = 9$ and $B = 1$ ($c,d$).

Figure 7 shows that $\psi_{C_T|\boldsymbol{b}}(\tau)$ and $\psi_{C_T|\boldsymbol{a}}(\tau)$ approximate $\psi_{T|\boldsymbol{b}}(\tau)$ and $\psi_{T|\boldsymbol{a}}(\tau)$ to within a scaling constant for these parameter values. Equivalently, we can approximate $C_T|\boldsymbol{b} \propto T|\boldsymbol{b}$ and $C_T|\boldsymbol{a} \propto T|\boldsymbol{a}$ when $r \approx 1$ and the partition sizes are small. That proportionality approximation is particularly accurate when the process fixes, starting from a small initial mutant population size ($b,d$). These results are analogous to those obtained for the fully connected Moran process [16]. The sojourn times of the fully connected Moran process do not significantly vary across state space when its population size is small, selection is weak, and the process fixes [16]. Therefore the scaling approximation $C_T|\boldsymbol{a} \propto T|\boldsymbol{a}$ is particularly accurate in this region of parameter space for the Moran process. However, if the Moran process achieves extinction, its sojourn times can vary significantly over state space, and the approximation $C_T|\boldsymbol{b} \propto T|\boldsymbol{b}$ may be inaccurate. Figure 7 shows that these observations hold for the complete bipartite graph when the graph is almost isothermal ($a,b$). When the partition sizes are highly asymmetric ($c,d$), both scaling approximations $C_T|\boldsymbol{a} \propto T|\boldsymbol{a}$ and $C_T|\boldsymbol{b} \propto T|\boldsymbol{b}$ are very accurate. This result suggests that the sojourn times of the complete bipartite graph do not vary significantly over state space if its partition sizes are highly asymmetric, e.g. the star graph.

Figure 8 is identical to figure 7, except we set $r = 3$. In the fully connected Moran process, the proportionality approximation $C_T|\boldsymbol{a} \propto T|\boldsymbol{a}$ loses accuracy as selection departs from $r \approx 1$ [16]. Figure 8 suggests that this result also holds for the complete bipartite graph ($b,d$). Given an appropriate scaling constant, we can accurately estimate the first few moments of $T|\boldsymbol{a}$ from $C_T|\boldsymbol{a}$, but higher-order moments are less accurately approximated than they were in figure 7. Figure 8 also suggests that our proportionality approximation is more accurate when the partition sizes are asymmetric (cf. $a,b$ and $c,d$),

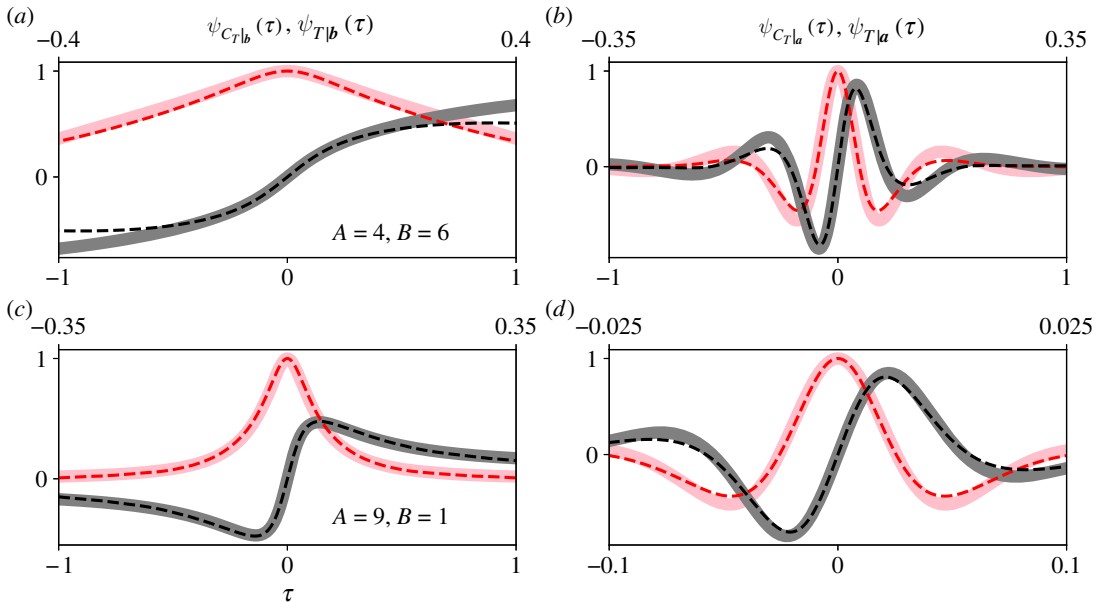

**Figure 8.** When selection is not weak, our proportionality approximation of $T|a$ from $C_T|a$ loses accuracy, particularly for higher moments. We set $r = 3$ in these plots; all other parameters are those stated for figure 7. Our approximation $C_T|b \propto T|b$ (a,c) appear to be as accurate as they were in figure 7. But our approximation $C_T|a \propto T|a$ (b,d) has noticeably lost accuracy, particularly for higher moments. Again, we observe that our proportionality approximation is more accurate when partition sizes are asymmetric (c,d). This observation suggests that the sojourn times of the complete bipartite graph do not appreciably vary over state space when partition sizes are asymmetric, e.g. the star graph.

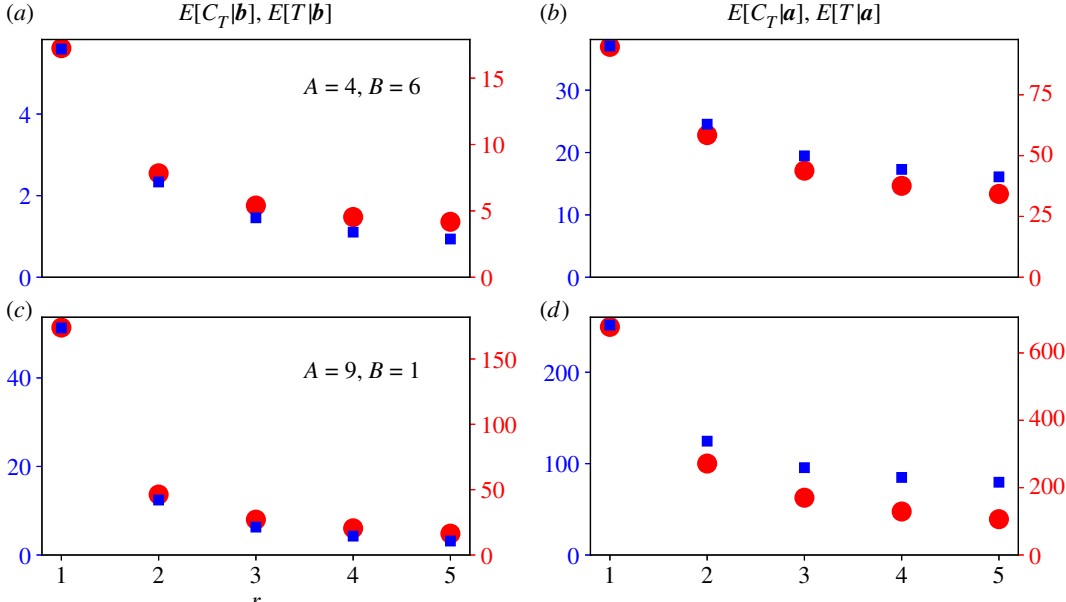

**Figure 9.** The relationship between the conditional means of $C_T$ and $T$ is complicated. We ran 100 000 simulations of the Moran process for five values of $r$, and for two bipartite graphs with the stated partition sizes $A$ and $B$ (a,b and c,d). We calculated the conditional means of $C_T$ (blue squares) and $T$ (red circles) of those simulations, conditional on extinction (a,c) or fixation (b,d). Blue squares are plotted against the left blue y-axis, and red circles are plotted against the right red y-axis. If the conditional means of $C_T$ and $T$ were proportional to each other, then the red and blue data points would perfectly overlap. But the relationship between $C_T$ and $T$ varies over transient states of the Moran process, which is why its conditional time distributions are hard to calculate. That relationship also depends on $r$, the bipartite graph's partition sizes, and whether the mutant population fixes or goes extinct. In all plots, $S_{a,0} = 1$ and $S_{b,0} = 0$.

as we observed in figure 7. Again, this observation suggests that the sojourn times of the complete bipartite graph do not appreciably change over state space when the partition sizes are asymmetric.

Figure 9 compares the conditional means $\mathbb{E}[C_T|b]$ and $\mathbb{E}[C_T|a]$ (blue squares) with $\mathbb{E}[T|b]$ and $\mathbb{E}[T|a]$ (red circles). We set $r = 1, 2, 3, 4$ and $5$, and ran 100 000 simulations of the Moran process on a bipartite

graph for each value of $r$. We calculated the means of those 100 000 simulations conditional on extinction ($a$,$c$) or fixation ($b$,$d$). We repeated these simulations for two bipartite graphs. The first bipartite graph had partition sizes $A = 4$ and $B = 6$ (figure 9$a$,$b$), and the second had partition sizes $A = 9$ and $B = 1$ (figure 9$c$, $d$). Each panel has two $y$-axes. The left, blue $y$-axis corresponds to the conditional means of $C_T$, and the right, red $y$-axis corresponds to the conditional means of $T$.

Figure 9 suggests that the relationship between the conditional means of $C_T$ and $T$ is not straightforward. When the mutant population size of the Moran process changes from $S_{t-1}$ to $S_t$, it remains in the state $S_t$ for a geometrically distributed number of time steps before changing again. If that geometric distribution was constant over all transient states of the process, then the conditional means of $C_T$ and $T$ would be proportional to each other. The proportionality constant would be that geometric distribution's mean, and the red and blue markers in figure 9 would overlap perfectly. However, the means of those geometric distributions depend on $S_t$. For example, if the mutant population is very close to extinction or fixation, then the probability of the mutant population size changing on a time step is small. On most time steps we will observe a resident replacing a resident or a mutant replacing a mutant. But if the mutant and resident population sizes are equal, then we are more likely to observe a change in the mutant population size on a time step. The state dependence of those geometric means is why the conditional absorption time distributions of the Moran process are so difficult to calculate [16]. Our martingale methodology shows that when we eliminate those state-dependent geometric distributions by focusing on 'active steps,' we obtain clean and exact expressions for that quantity's CCFs.

## 3. Discussion

Martingales can be interpreted as conservation laws for stochastic processes [16]. A martingale states that the expectation of some quantity does not change throughout a stochastic process. So if we know that expectation at the beginning of a stochastic process, then by induction we know it upon absorption. For example, we found two pairs of *state-independent* functions ($f_1(h)$, $g_1(h)$) and ($f_2(h)$, $g_2(h)$) such that equation (2.4) is true for the complete bipartite graph. Since we found a conservation law, we do not need to construct a Markov transition matrix [12,24], or evaluate recursion relations over all state space [13,37], or assume simplifying limits [14,37] to analyse the complete bipartite graph. Those alternative approaches are valuable because they are flexible tools to investigate a wider class of evolutionary graphs than we consider here. But if we can find a conservation law for a particular graph, then we can immediately exploit it to obtain elegant expressions for statistics of interest upon absorption.

The key step in applying martingale analysis is to somehow eliminate state dependence from an expectation that depends on some random variables of interest. This elimination step is easier for some random variables than others. For example, we have not yet found a martingale that depends on the number of time steps before absorption $T$ for the complete bipartite graph. But we can find a martingale that depends on the number of 'active steps' of the graph, $C_T$ [29,38]. So by switching our random variable of interest, we facilitate clean analysis. Martingales may not be applicable to all problems of interest in evolutionary graph theory. But they are very helpful in identifying problems that are conducive to tractable analysis.

The evolutionary graph theory literature has primarily studied the conditional distributions or moments of $T$ instead of $C_T$ [14,24,25,29,37]. We suggest switching their order of importance for four reasons. First, eliminating time steps where the graph does not change has no impact on the Moran process. By definition, the transition probabilities of the Moran process are unaffected by time steps where the mutant population remains unaltered [8,11]. Second, eliminating those time steps in simulations expedites computation time and reduces power consumption [38], especially as the mutant population size approaches extinction or fixation [17]. We can eliminate those time steps in simulations by calculating transition probabilities conditional on the mutant population changing, i.e. $\Pr(\mathbf{X_t}|\mathbf{X_t} \neq 0)$. Third, if we insist on obtaining conditional distributions of $T$, then we can sometimes closely approximate them from our conditional distributions of $C_T$ anyway [16]. Fourth, eliminating those time steps facilitates clean, elegant and exact expressions for the CCFs of active steps.

Our expressions for CCFs are exact because the Moran process must absorb exactly on either the fixation or extinction boundary. Since those boundaries are integers, and since the Moran process can only increase or decrease by 1, it cannot exceed them. Generally, random walks can exceed their absorbing boundaries. For example, consider a stochastic process where we continue adding

observations of standard Gaussian random variables until the cumulative sum exceeds one of two (constant) absorbing boundaries [28]. This stochastic process can absorb at an infinite number of possible values because it can exceed its boundaries. Martingales only provide approximate results for global statistics such as absorption probabilities and times when barrier excess is not zero [28] or otherwise calculable [39]. Determining when martingales provide accurate approximations of global statistics for such stochastic processes, or bounds on them, is an active research topic. But this issue is irrelevant for the Moran process [21,28,31,32] and related birth–death processes [7,15], because martingales yield exact results for these problems.

Martingales are a particularly powerful approach to study evolutionary graphs because they are exempt from the curse of dimensionality. As the dimensionality of a graph increases (i.e. as we divide a population into more partitions), martingale analysis does not necessarily increase in complexity. Previous results have demonstrated this remarkable property by calculating fixation probabilities for certain kinds of evolutionary graphs with arbitrary dimensionality [32]. Our results here suggest that we can extend our analysis to find the CCFs of $C_T$ for higher-dimensional graphs. We can obtain CCFs of $C_T$ for a one-dimensional graph (i.e. the fully connected Moran process [16]), and a two-dimensional graph (i.e. the complete bipartite graph). Therefore, we should be able to obtain them for graphs of arbitrary dimensionality as well.

Martingales' ability to scale with dimensionality is unmatched by other popular approaches to analysing such graphs, e.g. Markov chains and simulations [12,14,24,37]. As the dimensionality of a graph increases, the dimensionality of the Markov chain must increase because we need to account for more possible transitions of the graph on a time step. Calculating global statistics from a high-dimensional Markov matrix quickly becomes intractable, even if the elements in the matrix have simple mathematical forms [18]. Simulations quickly become prohibitively time-consuming to execute as graphs become more complex. When graphs have more partitions, they have more parameters (e.g. the population size in each partition is a parameter). Exploring how global statistics vary in high-dimensional parameter space is infeasible because simulation results are only valid for the specific parameter values we used in the simulation. Martingales can yield compact expressions for those global statistics that are valid over all parameter space, regardless of their dimensionality.

Some evolutionary graphs are probably not conducive to martingale analysis. We found a martingale by exploiting a symmetry in the state dependence of the mutant population increasing by one, and the resident population decreasing by one. For example, the state dependence in the transition probability of a mutant offspring from partition $A$ replacing a resident in $B$ is $S_{a,t-1}(B - S_{b,t-1})$. The state dependence in the transition probability of a resident offspring from partition $B$ replacing a mutant in $A$ is $(B - S_{b,t-1})S_{a,t-1}$. Since those state dependencies are the same, we can cancel them in equation (2.4). If we consider graphs with directed edges [29], then this symmetry is destroyed. It will be significantly more difficult, if not impossible, to cancel state dependencies in transition probabilities over all state space. So it will be very difficult to find a conservation law for an evolutionary graph with directed edges. Martingale analysis may be unsuitable graphs with directed connections.

The applicability of martingale analysis is also sensitive to whether birth or death occurs first in a birth–death process (i.e. a birth–death or death–birth process [1,15,40–42]), and whether the selection of the dying or reproducing individual is fitness-dependent. In the original Moran process, we select the reproducing individual before the dying individual, and birth selection is fitness-dependent. We showed that we can eliminate state dependence in evaluating equation (2.4) for the original Moran process on a complete bipartite graph. Now say we select an individual to die before choosing another to reproduce on a time step. If we define death selection to be fitness-dependent in the death–birth process, then we preserve the symmetry of state-dependence and martingale analysis remains applicable. But if the dying individual is chosen first, and reproduction selection is fitness-dependent, then that symmetry is destroyed. Seemingly trivial changes in the definition of the birth–death process can significantly impact the application of martingale analysis.

Martingales may also be applicable to other extensions of the Moran process constrained by graphs. Instead of haploid reproduction, we can consider diploid reproduction models, where two individuals can sexually reproduce only if they are connected by an edge on the graph [7]. We can consider birth–death processes with more than two competing species, each with different fitnesses [43,44]. We can consider heterogeneous graphs, where fitness is attributed to nodes on the graph, as well as the species of the individual occupying it [45]. We can consider evolutionary games on graphs, where individuals connected by graph edges compete in games for some pay-off [25,42,46–48]. Whether or not martingale analysis is applicable to any of these Moran process extensions is an open research question. To show that it is, we need to find a quantity whose expectation is either one (a product

martingale) or zero (a sum martingale), *regardless of the state of the process*. Then we can manipulate that conservation law to extract statistical quantities of interest. Finding such an expectation may be quite laborious or impossible, depending on the complexity of the stochastic process and the exploitable symmetries in it. In such cases, we should defer to other methods of analysis such as simulations, Markov chains, diffusion approximations, etc. But if we can find such an expectation, then martingale theory yields clean, elegant, exact and explicit expressions for statistics of interest. So those other methodologies should not be our default approaches to analysing evolutionary models, but rather our fallback options.

Data accessibility. Data and relevant code for this research work are stored in GitHub: https://github.com/travismonk/bipartite and have been archived within the Zenodo repository: https://doi.org/10.5281/zenodo.5504342.
Authors' contributions. T.M. conceived the study, did the maths, produced the figures, wrote the simulation code and drafted the manuscript. A.v.S. interpreted the mathematical results, checked the code and critically revised the manuscript.
Competing interests. The authors declare no competing interests.
Funding. No funding has been received for this article.

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
