## [Peer Review File · Royal Society Open Science]

Review History

RSOS-210657.R0 (Original submission)

Review form: Reviewer 1

Is the manuscript scientifically sound in its present form?

Yes

Are the interpretations and conclusions justified by the results?

Yes

Is the language acceptable?

Yes

Do you have any ethical concerns with this paper?

No

Have you any concerns about statistical analyses in this paper?

No

Recommendation?

Accept as is

Comments to the Author(s)

This is a well written manuscript that develops a theory of conditional characteristic functions and applies it to get formulas for the number of times that the mutant population size changes before going extinct or fixing in a bipartite graphs.

The paper can be accepted as is but the authors may consider adding some comparisons relating their results to the mean fixation or absorption times (such as in the form of figures for one or more bipartite graphs with mutant fitness on x axis and different times on the y axis, perhaps after an appropriate transformation)

Review form: Reviewer 2

Is the manuscript scientifically sound in its present form?

Yes

Are the interpretations and conclusions justified by the results?

Yes

Is the language acceptable?

Yes

Do you have any ethical concerns with this paper?

No

Have you any concerns about statistical analyses in this paper?

No

Recommendation?

Accept with minor revision (please list in comments)

Comments to the Author(s)

Please see attached report file (Appendix A).

Decision letter (RSOS-210657.R0)

Dear Dr Monk

On behalf of the Editors, we are pleased to inform you that your Manuscript RSOS-210657 "Martingales and the characteristic functions of absorption time on bipartite graphs" has been accepted for publication in Royal Society Open Science subject to minor revision in accordance

with the referees' reports. Please find the referees' comments along with any feedback from the Editors below my signature.

Please submit your revised manuscript and required files (see below) no later than 7 days from today's (ie 07-Sep-2021) date. Note: the ScholarOne system will 'lock' if submission of the revision is attempted 7 or more days after the deadline. If you do not think you will be able to meet this deadline please contact the editorial office immediately.

on behalf of Professor Andreas Kyprianou (Associate Editor) and Mark Chaplain (Subject Editor)
openscience@royalsociety.org

Associate Editor Comments to Author (Professor Andreas Kyprianou):

Associate Editor: 1

Comments to the Author:

The two referees were both positive about the contribution. There are some minor things that need thinking about that both referee's suggest. I don't see a need to go through another round of refereeing, so I would simply ask that you take your time in making the corrections before submitting the final files for publication.

Reviewer comments to Author:

Reviewer: 1

Comments to the Author(s)

This is a well written manuscript that develops a theory of conditional characteristic functions and applies it to get formulas for the number of times that the mutant population size changes before going extinct or fixing in a bipartite graphs.

The paper can be accepted as is but the authors may consider adding some comparisons relating their results to the mean fixation or absorption times (such as in the form of figures for one or more bipartite graphs with mutant fitness on x axis and different times on the y axis, perhaps after an appropriate transformation)

Reviewer: 2

Comments to the Author(s)

Please see attached report file.

===PREPARING YOUR MANUSCRIPT===

===PREPARING YOUR REVISION IN SCHOLARONE===

Author's Response to Decision Letter for (RSOS-210657.R0)

See Appendix B.

Decision letter (RSOS-210657.R1)

Dear Dr Monk,

I am pleased to inform you that your manuscript entitled "Martingales and the characteristic functions of absorption time on bipartite graphs" is now accepted for publication in Royal Society Open Science.

on behalf of Professor Andreas Kyprianou (Associate Editor) and Mark Chaplain (Subject Editor)
openscience@royalsociety.org

Appendix A

REFEREE REPORT ON RSOS-210657
“Martingales and the characteristic functions of
absorption time on bipartite graphs”
by T. Monk & A. van Schaik

Date of report August 17, 2021

1. PAPER SUMMARY

The paper investigates the Moran model on bipartite graphs and develops a method to explicitly compute the conditional characteristic functions

$$\mathbb{E}[\exp(i\lambda C_T)|\text{fixation at } T], \quad \mathbb{E}[\exp(i\lambda C_T)|\text{extinction at } T],$$

where T is the time at which the mutants become either extinct or comprise the whole population, and where $C_t, t = 1, 2, \dots$ counts the number of instants up to time t at which the size of the mutant population changes. The resulting characteristic functions are then discussed for several parameter values and compared to numerically obtained values for the characteristic function of T .

2. GENERAL REMARKS

The paper is original and mathematically sound. It extends previous works by the authors on how to exploit martingale techniques in a clever way to study fixation probabilities and fixation times in the Moran process. Their method is not only mathematically appealing but also of practical value, as it provides an analytical understanding of quantities which have hitherto only been numerically studied. The only drawback of the approach is that it relies heavily on symmetries in the specific example considered and it is dubious whether the method can be adapted for graphs which are not either bipartite or complete. The presentation of the paper could be a bit more polished in some places, I have provided a list of pointers below.

My recommendation is that the paper be accepted for publication after minor corrections.

3. SOME COMMENTS ON THE TEXT

- p.1, l.50 The first sentence is somewhat misleading, as it is not the (sole) purpose of stochastic processes to “model the spread of some novelty in a population”. How about *The spread of some novelty in a population can be modelled by stochastic processes* ?
- p.2, l.39/40 Instead of “the Moran process on evolutionary graphs” write *the Moran process on **more general** evolutionary graphs* or even just *the Moran process on **more general** graphs*. The latter suggestion applies to many instances throughout the paper, the graphs discussed in evolutionary graph theory are not evolutionary per se,

- but become “evolutionary graphs” by being used in the context of evolution models.
- p.2, last line Replace “complete” by something like *more detailed*.
- p.3, l. 41 It is not the graph which can be considered as a bivariate random walk, but the process on it.
- p.3, l. 56 Writing $\Pr(T|\cdot)$ for the conditional distribution clashes with the use of $\Pr(\cdot)$ as shorthand for ‘probability of’. It would be better to use e.g. $(\Pr(T = t|\cdot)_{t=0}^\infty)$.
- p.4, eq (21) The variable h is throughout a complex number, it would help if that was mentioned at its introduction. E.g. use *free complex variable* instead of just “free variable”.
- p.4, l.21 Calling (21) a martingale is semantically problematic. Either call it *product martingale property* or *product martingale relation*, or say that $h, f(h), g(h)$ satisfying (21) define a product martingale.
- p.5, l.10 “Assume that f and g are convex such that they have two possible complex values in the neighbourhood about $\tau = 0$.” This sentence is very hard to understand, the general reference [26] you mention does not help very much either. It becomes somewhat clearer what you mean here only in the next section, when given h you solve for $f(h), g(h)$. It seems you are referring to $x = f(h), y = g(h)$ as valid solutions to the martingale equations for given h ; f and g are then the implicit solution curves, but the connection to convexity of f, g in h is not apparent to me.
- p.8, Figure 3 In the top right panel the legend reads “ $\alpha \ 0.889 \approx 0.889$ ”.

Appendix B

Dr. Travis Monk,
International Centre for Neuromorphic Systems
The MARCS Institute
Western Sydney University
Sydney, Australia

internet www.westernsydney.edu.au/icns
e-mail
travis.monk@westernsydney.edu.au
telephone +61 2 4736 0668 (ext. 2668)
mailing address. Dr. Travis Monk
Western Sydney University

Bag 1797

Penrith, NSW, Australia 2751

Locked

Dear Prof. Kyprianou,

We thank you and the referees for your careful reading of our manuscript. We are pleased that our submission is of sufficient quality and interest to be published in *RSOS*. Please find below our responses to referees, and how we revised the manuscript.

Referee 1

Thank you for complimenting our manuscript's writing.

Comment 1: The authors may consider relating their results to the mean absorption times (such as in the form of figures for one or more bipartite graphs with mutant fitness on x axis and different times on the y axis).

Response: We added the requested figure as Fig. 9. We compare our results to mean absorption times for two bipartite graphs with mutant fitness on the x-axis and times on the y-axis. The final two paragraphs of the Results section discuss the new figure.

Referee 2

Thank you for noting the originality and mathematical rigour of our submission.

Comment 1: Referee 2 suggested 9 minor edits of the text to improve our presentation.

Response: We agree with all 9 suggested edits. We implemented them to improve our presentation. We highlighted them in red in our submitted document that marks all changes.

Again, we sincerely thank you and the referees for your time and expertise. We believe our manuscript has been improved by your diligence.

Sincerely,
Travis Monk and André van Schaik